# Sodium Alginate/β-Cyclodextrin Reinforced Carbon Nanotubes Hydrogel as Alternative Adsorbent for Nickel(II) Metal Ion Removal

**DOI:** 10.3390/polym14245524

**Published:** 2022-12-16

**Authors:** Aiza Farhani Zakaria, Sazlinda Kamaruzaman, Norizah Abdul Rahman, Noorfatimah Yahaya

**Affiliations:** 1Department of Chemistry, Faculty of Science, Universiti Putra Malaysia, Serdang 43400, Selangor, Malaysia; 2Natural Medicines and Product Research Laboratory (NaturMeds), Institute of Bioscience (IBS), Universiti Putra Malaysia, Serdang 43400, Selangor, Malaysia; 3Materials Processing and Technology Laboratory, Institute of Advanced Technology, Universiti Putra Malaysia, Serdang 43400, Selangor, Malaysia; 4Department of Toxicology, Advanced Medical and Dental Institute (AMDI), Universiti Sains Malaysia, Kepala Batas 13200, Penang, Malaysia

**Keywords:** water pollution, heavy metal ions, hybrid hydrogel, nickel(II) ion

## Abstract

Water pollution issues, particularly those caused by heavy metal ions, have been significantly growing. This paper combined biopolymers such as sodium alginate (SA) and β-cyclodextrin (β-CD) to improve adsorption performance with the help of calcium ion as the cross-linked agent. Moreover, the addition of carbon nanotubes (CNTs) into the hybrid hydrogel matrix was examined. The adsorption of nickel(II) was thoroughly compared between pristine sodium alginate/β-cyclodextrin (SA-β-CD) and sodium alginate/β-cyclodextrin immobilized carbon nanotubes (SA-β-CD/CNTs) hydrogel. Both hydrogels were characterized by attenuated total reflectance Fourier transform infrared spectroscopy (ATR-FTIR) spectral analysis, field emission scanning electron microscopy (FESEM), electron dispersive spectroscopy (EDX), thermogravimetric analysis (TGA) and Brunauer–Emmett–Teller (BET) surface area analysis. The results showed SA-β-CD/CNTs hydrogel exhibits excellent thermal stability, high specific surface area and large porosity compared with SA-β-CD hydrogel. Batch experiments were performed to study the effect of several adsorptive variables such as initial concentration, pH, contact time and temperature. The adsorption performance of the prepared SA-β-CD/CNTs hydrogel was comprehensively reported with maximum percentage removal of up to 79.86% for SA-β-CD/CNTs and 69.54% for SA-β-CD. The optimum adsorption conditions were reported when the concentration of Ni(II) solution was maintained at 100 ppm, pH 5, 303 K, and contacted for 120 min with a 1000 mg dosage. The Freundlich isotherm and pseudo-second order kinetic model are the best fits to describe the adsorption behavior. A thermodynamic study was also performed. The probable interaction mechanisms that enable the successful binding of Ni(II) on hydrogels, including electrostatic attraction, ion exchange, surface complexation, coordination binding and host–guest interaction between the cationic sites of Ni(II) on both SA-β-CD and SA-β-CD/CNTs hydrogel during the adsorption process, were discussed. The regeneration study also revealed the high efficiency of SA-β-CD/CNTs hydrogel on four successive cycles compared with SA-β-CD hydrogel. Therefore, this work signifies SA-β-CD/CNTs hydrogel has great potential to remove Ni(II) from an aqueous environment compared with SA-β-CD hydrogel.

## 1. Introduction

Heavy metals are common inorganic pollutants classified as having high-stability, bioaccumulative, toxic and persistent properties that do not naturally degrade into harmless end products under normal environmental conditions [1,2]. Ni(II) is one of the heavy metals that frequently contributes to water contamination of the main water reservoir which will be further discussed in this study, alongside the reduction of water pollution issues using adsorption techniques. Ni(II) possesses unique properties, such as excellent stability in chemical, thermal and surface activity; this has driven them to be ubiquitously utilized in many manufacturing industries to produce our commodities and goods, such as surface-finishing materials, fire-fighting appliances, batteries, plastics, kitchen utensils, fertilizers and pesticides [3,4]. However, strong chemical oxidation of Ni(II) is severely compromised and degraded by conventional nanotechnology methods, resulting in damage and severe effects that threaten biotic and abiotic ecosystems, even when exposed at significantly lower concentrations [5]. Although living organisms will also tend to present with seizure, spasm and diseases such as water sickness, nausea, dermatitis, skin ulceration, vomiting, kidney damage and anemia, the most worrying effect of Ni(II) is its potential to cause death [6,7,8]. Due to the circumstances mentioned above, Ni(II) remediation is still an unsettling issue which represents a significant concern among environmental scientists and researchers worldwide. Therefore, the fabrication of pre-eminent water treatment innovations with a strong affinity for tackling the spread of Ni(II) pollutants in the aqueous environment has been promulgated and explored in this research work [9].

An economical, innovative, green adsorbent based on various natural raw materials was prepared in order to overcome the drawbacks of the existing adsorbent. The following natural biopolymers are widely used as precursors due to their good reproducibility, biodegradability, easy accessibility, lower manufacturing regeneration cost and environmental friendliness [10]. Sodium alginate (SA) is a binary heteropolymer derived from brown algae or sargasso, consisting of a linear arrangement of blocks composed of 1,4-linked α-L-guluronic acid (G-block) and β-D-mannuronic acid (M-block) units in pyranose form, and was selected as a type of bioadsorption substrate [11,12]. It has been reported as a superior compound that can form a gel by crosslinking with divalent or polyvalent calcium ions to induce the formation of ionotropic metal–alginate complexes [11,12,13]. The utilization of sodium alginate makes the remediation process eco-friendlier because it is edible, biodegradable and has renewable properties that can be found abundantly in nature [14,15]. β-cyclodextrin (β-CD) is a porous macrocyclic sugar consisting of seven glucose units linked by α-(1,4) bonds [16]. β-CD is also considered an excellent adsorbent because it can improve the solubility and stability of functional materials. It is also a biodegradable, biocompatible, water-soluble and nontoxic polymer derived from chitin, which is abundant and readily available from the shell waste of crustaceans [17,18]. The most crucial property of β-CD is the formation of three-dimensional chiral cavities in the microenvironment which consist of a hydrophobic internal cavity and a hydrophilic external characteristic. In these cavities, the formation of host–guest and stable inclusion complexes with thousands of organic, inorganic, and biological micropollutant compounds is selectively allowed [18,19]. The abundance of hydroxyl and carboxyl functional groups on the molecular surface of β-CD voids enables non-covalent intermolecular interactions with the contaminants, and thus the incorporation of β-CD in hydrogel composites can increase their adsorption capacity to remove nickel metal ions [16,18,20,21]. Duman et al. stated in their article that β-CD is a good candidate for adsorbent preparation. However, the abundance of hydroxyl groups in β-CD influences water’s solubility during water treatment. Because of this, the authors propose that β-CD has to be modified by combining it with other materials using blending technique to improve its stability, insolubility and selectivity properties as an adsorbent [21].

In recent years, carbon nanotubes (CNTs) have been used as traditional carbon adsorbents implemented in several distinct contexts, such as electronics, biosensors, catalysis, and water treatment applications [22,23]. Furthermore, this carbon nanostructure has received appreciable attention in many water remediation applications due to its high specific surface area, unique hollow tubular structure, abundant active sites, superior isotropic properties, favorable penetration and excellent electrical and thermomechanical properties [24,25]. However, CNTs have some shortcomings: they are easy to aggregate and have poor recyclability and low dispersion characteristics, because the absence of the polar functional group on the surface of CNTs restricts their application in the adsorption process. In this regard, CNTs have been modified to impart polar functional groups on their surface and therefore enhance their performance [20]. Zhan and co-workers stated that the active sites of various negatively charged oxygen-containing groups (OH, COOH, and epoxy groups) on carbon nanotubes could significantly enhance the diffusivity of contaminants through electrostatic interaction and hydrophobic interaction between the targeted pollutants [26]. Therefore, improving the performance of CNTs was desired to solve the problem of separation and dispersion difficulty. CNTs were grafted onto alginate and β-CD to form a polymeric composite hydrogel adsorbent and significantly expand its application range to improve water pollution issues caused by various industries. In particular, the applicability of hydrogel as a class three-dimensional (3D) material has prompted interest due to its excellent physicochemical properties, such fast adsorption kinetics, eco-friendliness, low cost, great porosity and good recovery and reusability. In addition, the introduction of the CNTs into the polymeric hydrogel can further improve the individual components’ limited operational properties in terms of stability, selectivity, or sensitivity by combining their special recognitions and capabilities into a single functional hydrogel composite network [27,28]. For instance, a study conducted by Makhado et al., the incorporation of CNTs into poly(acrylic acid) and sodium alginate was successfully achieved through graft co-polymerization. The authors stated that the prepared hydrogel nanocomposite applied to the adsorption of dyes holds excellent potential for implementation in water treatment applications [29]. Mohammadinezhad et al. synthesized acrylamide and itaconic acid with the presence of multi-walled CNTs to investigate the removal of Pb (II) from an aqueous environment. The authors mentioned that the super hydrogel adsorbent containing CNTs showed better adsorption behavior, thermal stability and mechanical properties than hydrogel without CNTs [30]. Lin et al. fabricated an adsorbent that incorporated multi-walled CNTs covalently grafted with β-CD; there, the improvement of CNTs dispersion, specific surface area pore volume and thermal stability was observed. The authors reported that the thermal stability of magnetic composite hydrogel (β-CD@Fe_3_O_4_/MWCNT) was reduced only by about 8.7% compared with pure β-CD that saw 75.5% weight loss after degradation [20]. Based on the literature, we hypothesized that incorporating CNTs into the hydrogel network will enhance thermal stability, thermomechanical properties, and adsorption capacity in water remediation applications. Hence, a new sustainable adsorbent based on the combination of biopolymers, with an impressive number of binding sites for the adhesion of Ni(II), was developed.

In the present study, SA-β-CD/CNTs hydrogel beads were successfully synthesized by chemical grafting technique to select and effectively remove Ni(II) from the aqueous environment. The obtained composite SA-β-CD/CNTs hydrogel bead was characterized by FTIR, BET, TGA, FESEM, and EDXs analyses. Accordingly, the batch adsorption experiment process was examined by controlling several parameters, such as pH, dosage, contact time, initial concentration and temperature, in detail. In addition, the isotherms, kinetics, and thermodynamics of Ni(II) were studied. Finally, the application in the actual water environment and reusability of SA-β-CD/CNTs hydrogel beads were investigated.

## 2. Material and Methods

### 2.1. Material

Carboxylated multi-walled carbon nanotubes with an average diameter of about 9.5 nm were obtained from Sigma–Aldrich, (Darmstadt, Germany). Ni(II) solution was purchased from Scharlau, (Barcelona, Spain). Sodium alginate biopolymer with viscosity of 300–400 Pa s was purchased from Chemiz (M) Sdn. Bhd. (Selangor, Malaysia). β-cyclodextrin was purchased from Sigma–Aldrich, (Darmstadt, Germany). Calcium chloride was purchased from Progressive Scientific Sdn. Bhd. (Selangor, Malaysia). All other chemical reagents used were of chromatographic or analytical grade or better, and were used as received. 0.1 M NaOH and 0.1 M HCl were used to adjust the pH of the solutions. Distilled water was used to prepare all Ni(II) ion solutions.

### 2.2. SA-β-CD/CNTs Hydrogel Preparation

SA-β-CD/CNTs hydrogel beads were synthesized. CNTs were dispersed homogeneously for 2 h in distilled water using an ultrasonicator dispersion technique. Sodium alginate and β-cyclodextrin were added to the homogeneous mixture solution with a mass ratio of 1:1. The mixed solution was continuously stirred to form a homogeneous viscous solution. Then, SA-β-CD/CNTs gel beads were formed by adding them dropwise into a 5 wt% CaCl_2_ solution using a syringe at a rate of 6 mL/min. The hydrogel beads were left undisturbed in the CaCl_2_ solution for 48 h. All the hydrogel beads were filtered using a funnel and washed several times with distilled water to obtain stable SA-β-CD/CNTs hybrid hydrogel beads, and were air-dried for 1 h before use in the adsorption process.

### 2.3. Characterizations of SA-β-CD/CNTs Hydrogel

SA-β-CD/CNTs were characterized by several high-technology instruments. Fourier emission scanning electron microscopy (FESEM, JEOL JSM-7600F, Tokyo, Japan) operating at 15.0 kV was used to determine the surface morphology of the hydrogel. Before scanning, dried SA-β-CD/CNTs hydrogel beads were attached to double-sided adhesive tape which was made electrically conductive (10 mm stubs). FESEM images were obtained with excitation voltage of 5 kV and magnification varied from 35.0 to 25.0 kX. The elemental identification of SA-β-CD/CNTs hydrogel was performed by Energy Dispersive X-Ray spectrometry (EDXs). Attenuated total reflectance Fourier transform infrared spectroscopy (ATR-FTIR) spectrometer (Perkin Elmer Spectrum RXI, Waltham, MA, USA) was applied to identify the functional groups of SA-β-CD/CNTs gel bead. Thermogravimetric analysis (TGA/SDTA 851, Mettler Toledo, Greifensee, Switzerland) was conducted to determine the thermal stability of the SA-β-CD/CNTs gel beads and SA-β-CD/CNTs hydrogel beads produced from the research under N_2_ atmosphere at a heating rate of 10 °C/min from 25 °C to 600 °C.

TGA/SDTA 851.

### 2.4. Adsorption Experiment of Ni(II) Solution

The adsorption of Ni(II) on sodium alginate-immobilized β-cyclodextrin/carbon nanotube hydrogel beads was studied in batch mode system. The effect of operative variables, including pH adjustment, initial Ni(II) concentration, adsorbent dosage, contact time between the hydrogel beads and Ni(II) solution and temperature on adsorption performance, was evaluated. Ni(II) working solution was prepared from the appropriate stock solution by diluting with distilled water. 75 mL of the Ni(II) solution was placed in a fixed-volume conical flask. SA-β-CD/CNTs hydrogel beads were added to the solution with constant stirring at constant room temperature and shaken to establish equilibrium. Then, the adsorbent was separated through a filter and the equilibrium concentration, Ce, of the nickel(II) solution was measured using inductively coupled plasma atomic emission spectroscopy (ICP-AES) (Perkin Elmer Optima 2100 DV, Shelton, CT, USA).

The parameter of adsorption capacity (q_e_) and removal efficiency (%) of Ni(II) adsorption was calculated by using the following equation:(1)$$q_{e}=\frac{\mathrm{Co}-\mathrm{Ce}}{W}\;*\;V$$
(2)$$\mathrm{Removal}\;\mathrm{efficiency},\;\%=\frac{\mathrm{Co}-\mathrm{Ce}}{\mathrm{Co}}\;*\;100$$
where C_e_ and C_o_ are the initial and equilibrium concentration of metal ions in solution. V is the volume (L) and W is the weight (g) of the adsorbent, respectively. Isotherm modelling studies were accomplished by utilizing 75 mL of Ni(II) solution with SA-β-CD/CNTs hydrogel (1000 mg) for 45 min at 303, 313, and 323 K. The kinetic investigations were studied by contacting 75 mL of Ni(II) solution with SA-β-CD/CNTs hydrogel (1000 mg) for 45 min at 303 K.

#### 2.4.1. Adsorption Isotherm and Kinetics

To study the adsorption nature of Ni(II) onto the SA-β-CD/CNTs hydrogel, Langmuir and Freundlich isotherm models were adapted. Their expressions are as follows:1/q_e_ = 1/Q_max_ + 1/bQ_max_ . 1/C_e_
(3)
where C_e_ is the liquid-phase concentrations of Ni(II) at equilibrium expressed in mg/g, q_e_ is the amount of Ni(II) adsorbed at equilibrium expressed in mg/g, Q_max_ is the maximum monolayer adsorption capacity expressed in mg/g, and b is the Langmuir constant expressed in L/mg or L/mol. The values of Q_max_ and b were calculated from the slope and intercept of the different straight lines representing to different temperatures [31]. Freundlich isotherm expression can be defined as follows:
ln q_e_ = ln Kf + (1/n) ln C_e_(4)
where C_e_ is the liquid-phase adsorbate concentrations at equilibrium (mg/L) and q_e_ is the amount of solute adsorbed at equilibrium (mg/g). Kf and 1/n are Freundlich constants related to the capacity of sorption and favorability of sorption, respectively. Kf (mg/g) (L/mg)^1/n^ is the adsorption capacity of the adsorbent and n indicates the degree of favorability of an adsorption process. In addition, the sorption is considered favorable when the value of n in the range of 0 < n < 10. The greater the value of n, the more favorable the adsorption process [31,32].

To ascertain the adsorption mechanism of Ni(II) ions, pseudo-first order and pseudo-second order kinetic models were used. The adsorption kinetic study is usually used to identify the adsorption rate and potential rate-controlling of the adsorption mechanism process [33]. The experimental adsorption phenomenon ordinarily employs the pseudo-first order, which focuses on the difference in equilibrium adsorption capacity as well as the adsorbed amount; the pseudo-second order model, which considers the assumption of the rate-limiting step, involves chemisorption. In relation to that, the pseudo-first order and pseudo-second order have been defined as the following equation:ln (q_e_ − q_t_) = ln q_e_ − k_t_ t(5)
t/q_t_ = 1/k_2_ q_e_^2^ + 1/q_e_ . t(6)
where, q_e_ (mg/g) means equilibrium adsorption capacity, $q_{t}$ (mg/g) means the adsorption capacity at t moment, and k_1_ (1/min) and k_2_ (1/min) the rate constant for the pseudo-first order kinetic model and pseudo-second order kinetic model, respectively [31,34].

#### 2.4.2. Study of Adsorption Thermodynamic

(7)$$K_{d}=\frac{q_{e}}{C_{e}}$$(8)$$\ln\;K_{d}=-\Delta H/\mathrm{RT}+\Delta S/R$$ΔG = ΔH − TΔS (9)
where K_d_ (L/g) is the distribution coefficient; q_e_ (mg/g) is the adsorption capacity at equilibrium; C_e_ (mg/L) is the concentration of the Ni(II) solution at equilibrium; R (8.314 J/mol K) is the ideal gas constant; ΔH (kJ/mol) is the standard change in enthalpy; ΔS (J/mol K) is the standard change in entropy; and ΔG (kJ/mol) is the Gibbs free energy change in a given process.

### 2.5. Regeneration Studies

The regeneration experiments were performed consecutively in four adsorption cycles to evaluate the regeneration performance of the hydrogel. SA-β-CD/CNTs hydrogel with Ni(II) was eluted with 0.1 M HCl and was washed with distilled water to remove any residuals for further use. The final concentration was analyzed after the adsorption of Ni(II) solution.

### 2.6. Real Water Sample Experiment

For this study, the real environmental water samples were spiked with a known concentration of Ni(II) standard (100 mg/L) to obtain the particular Ni(II) metal pollutant solutions of 100 mg/L in the wastewater collected. The SA/β-CD/CNTs hybrid hydrogel adsorbent was tested for its adsorption capability of Ni(II) from the real wastewater sample. The optimum conditions adapted from all parameters were applied to evaluate the removal uptake of the adsorbent towards Ni(II) metal. The real wastewater sample pH was adjusted to ideal pH value (pH 5), the optimum dosage of SA/β-CD/CNTs hydrogel (1000 mg), a constant contact time (120 min), volume (75 mL), temperature (303 K) and a shaking speed of 200 rpm. The final concentration of the Ni(II) was determined using ICP-AES spectroscopy technique.

## 3. Results and Discussion

### 3.1. The Novel Aspect of This Work 

We acknowledged that there has been no work of literature that discusses the fabrication of calcium alginate/β-cyclodextrin hydrogel incorporated with carbon nanotubes. Figure 1 shows the physical image of the synthesized SA-β-CD hybrid hydrogel without (a,b) and with (c,d) CNTs. The difference between SA-β-CD and SA-β-CD/CNTs can be distinguished by the color of the beads. Figure 1a,b shows the white color of the SA-β-CD hybrid hydrogel bead, indicating the absence of the CNTs in the polymer matrix. Nonetheless, calcium alginate and β-cyclodextrin have been reported to be widely used individually for removing organic (methylene blue, dicholorophenol) and inorganic pollutants (heavy metals). There are reports about preparing hybrid hydrogel between calcium alginate and β-cyclodextrin composite for potential use in tissue engineering [35,36,37]. The authors mentioned that both of the materials contain various functional groups that could be beneficial to enhancing adsorption capacities. In this study, we will further focus on the process of preparing a smart adsorbent and discuss its adsorption behavior towards the removal of Ni(II) from aqueous environment samples. Sodium alginate, β-CD, and MWCNTs were successfully prepared through graft co-polymerization when Ca^2+^ involved the ion-exchange reaction to form a single functionalized hydrogel matrix. The probable schematic interactions in the preparation of adsorbent are depicted in Figure 15. The abundance of H-bond donor in β-CD supports the formation of the hydrogen bonding with alginate and CNTs. In addition, other oxygenated functional groups, such as carboxyl and hydroxyl, belonging to alginate and β-CD, were also liable to interact with each other during the immobilization process. Furthermore, as MWCNTs are rich in oxygenated functional groups that face outward, they can easily and efficiently crosslink with β-CD through acetalization [38]. The addition of CNTs onto the hydrogel structure was expected to overcome the collapsing and high solubility problems caused by the weakening of electron repulsion and the reduction of oxygenated functional groups during the sorption process [38,39]. In addition, the concentration of the Ni(II) solution after and before adsorption was used to study the change in Ni(II) concentration in the solution. Moreover, several adsorption conditions, including initial concentration, pH, and contact time were thoroughly directed to ascertain the adsorption phenomenon between SA-β-CD/CNTs hybrid hydrogel beads and Ni(II) removal. On top of that, composite hydrogel may make advances as an alternative adsorbent material due to its satisfactory behavior, large porosity, high surface area, and extensive surface reactivity, which denoted suitability for use in water remediation.

### 3.2. Spectroscopic Study of SA-β-CD/CNTs Hydrogel Beads

The components of SA-β-CD/CNTs hydrogel beads have been determined by FTIR technique to observe the functional groups in the structure. The infrared spectra of three successful types of hydrogels are compared in Figure 2. All distinctive absorption bands of SA, SA-β-CD and SA-β-CD/CNTs were observed in the similar expected regions, but with slight differences in the peak positions, peak intensities and peak shapes. A broad peak can be observed at 3394, 3395, 3424 cm^−1^, respectively, indicating OH stretching vibrations in the hydroxyl group for all hydrogels [24,40]. In the spectrum of SA, a broad peak at 1030 cm^−1^ emanates from the stretching of C-O group [40,41]. The presence of the arabinosyl functionality was verified by the presence of a sharp peak at 1261 cm^−1^, while the other sharp peaks at 940 cm^−1^ and 874 cm^−1^ are attributed to the β-glycosidic linkage between the mannuronic and guluronic units in the G, M unit group stretching vibration of SA [15,42]. Meanwhile, the broad peaks at 2963 cm^−1^ and 1432 cm^−1^ were due to the sp^3^ C-H asymmetric stretching mode and the C-OH bending vibration, respectively [43,44]. The broad peaks at 3394 cm^−1^ and 1027 cm^−1^ are due to the stretching vibration of O-H and C-O-C groups [45]. After graft polymerization of sodium alginate and β-CD, an obvious peak intensity at 1728 cm^−1^ presented. The peak suggests that β-CD groups were successfully grafted onto the SA backbone [46]. The sharp peak at 1632 cm^−1^ and 1728 cm ^−1^ is assigned to symmetric and asymmetric vibrations of the COO^−^ groups [46,47]. However, the absorption intensity of the carboxyl group after the addition of CNTs was obviously decreased and shifted to 1641 cm^−1^, indicating most of the carboxyl group had been successfully cross-linked with the hydroxyl group of the SA and β-CD in the system [44]. Meanwhile, the peak at approximately 1730 cm^−1^ in the spectrum obtained for the SA-β-CD/CNTs hydrogel became more intense compared with the peak at 1725 cm^−1^ in the SA and SA-β-CD hydrogel spectra, indicating that more COO^−^ groups exist after the addition of CNT [48]. In contrast, the peaks located at 1432 cm^−1^ attributed to C-OH disappeared after the incorporation of CNT in the SA-β-CD hydrogel. The peak characteristic of SA-β-CD located at 3395 cm^−1^ reduced and broadened to 3423 cm^−1^ after the addition of CNTs, with different O-H stretching vibration intensities showing an increase the number of O-H forms present due to internal hydrogen bonding between carboxyl groups [29,48]. The absorption bands at 1227 cm^−1^ and 1035 cm^−1^ correspond to C-O-C stretching and C-O stretching vibration bonds, respectively [20,49]. The increase in peak intensity was expected to occur due to the electrostatic interaction of the carboxyl group of CNTs with SA and β-CD to form ester group [29,50].

After the adsorption of Ni(II), the FTIR absorption bands’ intensity usually shifted to lower and higher wavenumbers. The broad absorption peak represents that the stretching vibration of O-H has broadened and shifted to 3387 cm^−1^. An intensive peak at 1736 and 1641 cm^−1^ were combined to become a low absorption peak, and a low peak at 1639 cm^−1^ indicates COO^−^ became involved to interact with Ni(II) ions. The adsorption of Ni(II) could be further confirmed when the peak at 1227 cm^−1^ disappeared, suggesting the C-O to hydroxyl group has been replaced by Ni(II) ion. The results above concluded that OH and COOH groups that participated in Ni(II) ions adsorption onto SA-β-CD/CNTs hydrogel aligned with work published by Antić et al., and Kavitha et al. [50,51].

### 3.3. Morphological Study of SA-β-CD/CNTs

FESEM micrographs of the SA-β-CD/CNTs hydrogel bead adsorbent before the adsorption process can be seen in Figure 3. Figure 3 shows the images of hydrogel beads before adsorption at different magnifications. Figure 3a shows a well-defined structure that is reliable for use as an adsorbent. Figure 3b shows a rough and porous structure, indicating that it is suitable for use in effective adsorption. The non-smoothness and many wrinkles on the hydrogel surface indicate the presence of many functional groups and more active adsorption sites on the material, thereby showing a triple network structure between SA, β-CD, and CNTs has been successfully achieved [52].

Furthermore, BET analyses were carried out to determine the surface area and pore size of the SA-β-CD and SA-β-CD/CNTs hydrogel beads. The results prove that the surface area, with a 1:1 ratio of SA-β-CD hydrogel beads and the incorporated CNTs, has improved when compared with SA-β-CD and the absent CNTs. The surface area of SA-β-CD hydrogel containing CNTs has been reported to be 27.2424 m^2^/g, which was improved twice compared with SA-β-CD hydrogel (12.6018 m^2^/g) and the pure alginate reported by Cataldo et al., (about 0.17 m^2^/g) [53]. Using IUPAC classification, it exhibited a type III isotherm with an H3 hysteresis loop. It has a mesoporous structure, because the mean of the pore size diameter was 4.9 nm; the pore size of SA-β-CD was 4.7 nm, which is within the range of mesoporous material (2–50 nm). Thus, both SA-β-CD and SA-β-CD/CNTs hydrogel have been proven suitabe for nickel’s penetration site, because the nickel’s molecular size is only 0.124 nm. Nonetheless, SA-β-CD/CNTs hydrogel beads were expected to have higher adsorption capacity for Ni(II), which will be further discussed in the subsequent investigation.

### 3.4. Elemental Composition of SA-β-CD/CNTs Hydrogel

An energy dispersive X-Ray spectrometry (EDXs) analysis was employed to study the elemental identification of the SA-β-CD/CNTs hydrogel. Figure 4 shows the EDXs spectrum of the SA-β-CD/CNTs hydrogel, showing the presence of Ca peak and the resulting composition of Ca ion with an average of 14.96%. Based on this result, it was confirmed that Ca ions accumulated in the SA-β-CD/CNTs hydrogel, which is consistent with previous studies conducted by Asadi et al. [54], which showed that a successful ionic crosslinking reaction occurred between negative carboxylate groups of the alginate ring and calcium ions during the formation of hydrogel beads [12].

### 3.5. Thermal Stability Analysis

Figure 5 depicts a thermogram of SA-β-CD and SA-β-CD/CNTs hydrogel adsorbent studied for thermal stability. The first degradation curve indicates the water evaporation for the SA-β-CD peak occurred at 80 °C to 100 °C. The second weight-loss stage was pronounced from 125 °C to 200 °C; it is due to the decomposition of the egg-box structure formation from the ion exchange between calcium ions and sodium ions of the alginate [49]. Moreover, the major degradation peak of SA-β-CD and SA-β-CD/CNTs hydrogel can be observed when the temperature rises from 200 °C to 34 °C, corresponding to the decomposition of the hexatomic ring chain structure and oxygenated functional group [44,55]. The thermogram shows that the curve trends of SA-β-CD and SA-β-CD/CNTs hydrogel were similar in the temperature range from 80 °C to 100 °C, mainly caused by the loss of interstitial water [56,57]. The degradation peaks of both hydrogels after heating at 340 °C show the residue remaining found to be 32.7977% and 36.3786% for SA-β-CD and SA-β-CD/CNTs, respectively. This weight loss of SA-β-CD/CNTs is lower compared with SA-β-CD because the addition of CNTs led to an increase in the surface area and enhance the stability of the SA-β-CD/CNTs hydrogel [58]. The peak belongs to SA-β-CD/CNTs was shifted to a high degradation temperature, demonstrating their excellent stability [59]. The temperature difference might be attributed to the incorporation of CNTs in the polymer matrix, which influences the hydrogel’s stability, toughness and stiffness against thermal treatment [60]. Compared with the double polymer matric of SA-β-CD hydrogel, SA-β-CD/CNTs improved thermal stability in the triple-network by incorporating carbon nanomaterials.

### 3.6. Batch Adsorption Studies

In the study of adsorption properties, Ni(II) was used as the model contaminant in an aqueous environment. Several operational parameters were investigated through the batch system technique to ascertain the optimum condition that influenced the adsorption process, such as adsorbent dosage, pH, contact time and temperature. Prior to the investigation of the adsorption operational parameters, the optimal number of CNTs in the adsorbent was determined to assess their efficacy in a substantial adsorption capacity. This study was performed using 75 mL of Ni(II) solution, and the adsorption capacities of each parameter were calculated using Equation (1). The adsorbent dosage was examined from 200 to 1000 mg at pH neutral. Next, several initial concentrations of Ni(II) solution were studied, from 25 mg/L to 100 mg/L; pH was studied from 2 to 6, the contact time was studied from 0 min to 120 min, and the temperature conditions were studied from 293 K to 333 K, using 1000 mg of the optimized amount of adsorbent. Based on the optimization result, the ideal Ni(II) adsorption condition was found at a pH of 5, at temperature of 313 K, a contact time of 45 min and a finest adsorbent dosage of 1000 mg. The result of these studies is shown in Figure 6, Figure 7, Figure 8, Figure 9, Figure 10, Figure 11, Figure 12, Figure 13 and Figure 14.

#### 3.6.1. Effect Mass of Incorporated CNTs

The impact of CNTs on the removal efficacy of Ni(II) metal with varying numbers (0 mg, 1 mg, 5 mg and 10 mg) of CNTs has been studied. In this study, SA-β-CD/CNTs hydrogel is denoted as SA-β-CD, SA-β-CD/1 mg-CNTs, SA-β-CD/5 mg-CNTs and SA-β-CD/10 mg-CNTs, respectively, for better understanding. Figure 6 shows that the percentage removal of the Ni(II) improved after incorporating CNTs into the hydrogel–polymer matrix. Comparison of the percentages of removal between SA-β-CD hydrogel and SA-β-CD/CNTs hydrogel shows that Ni(II) is more effectively eliminated by SA-β-CD/CNTs, in which more direct proportionality indicates higher adsorption capacity. Nonetheless, when the CNTs’ concentration increased, the graph below showed a lower adsorption capacity in removing Ni(II) metal than the SA-β-CD/CNTs hydrogel containing low numbers of CNTs (1 mg). It shows that the SA-β-CD/CNTs hydrogel adsorbent with 1 mg CNTs has the highest and most significant capacity to remove Ni(II) metal. This may be because of its stable structure and rigid physical properties compared with those with higher numbers of CNTs [52,61]. In addition, a further increase in CNTs’ concentration may have caused domination of the mutual interaction forces between CNTs; it probably enhanced the aggregation of the CNTs and, thus, influenced them to stick together. For this reason, the exposure of the CNTs surfaces’ active adsorption sites in SA-β-CD/CNTs hydrogel to Ni(II) is reduced; this is similar to the finding of Hosseinzadeh [62]. Therefore, SA-β-CD/1 mg-CNTs has been considered the optimum mixture used for the following operational parameters in this study.

#### 3.6.2. Effect of SA/β-CD/CNT Hydrogel Dosage Effect

The dosage is a prime factor influencing the adsorption capacity of Ni(II) metal on SA-β-CD/CNTs hydrogel. Figure 7 indicates the percentage removal of SA-β-CD/CNTs hydrogel towards Ni(II) metal from 200 mg to 1000 mg, with the other parameters kept constant. The slope shows that the removal percentage gradually increased when the amount of SA-β-CD and SA-β-CD/CNTs hydrogel increased. The maximum percentage reported was up to 43.56% and 52.76% using 1000 mg of SA-β-CD and SA-β-CD/CNTs hybrid adsorbent, respectively. The reason for this is that increasing the dosage mass possibly enhanced the number of functional groups and provided a large surface area for the adsorbent that can interact with Ni(II) metal in both SA-β-CD and SA-β-CD/CNTs hydrogel adsorbent. In addition, the higher dosage indicates a greater affinity between the sorbent surface and Ni(II) metal, which then affects the removal of ions [63]. Moreover, when comparing the percentage removal towards both adsorbents, SA-β-CD/CNTs hydrogel was reported to remove more Ni(II) than SA-β-CD due to the presence of the oxygenated functional groups of CNTs in the adsorbent, which thus extensively influence the removal uptake of Ni(II). Therefore, 1000 mg of SA-β-CD/CNTs hydrogel was used as the optimum dosage for the subsequent adsorption study.

#### 3.6.3. Effect of pH

The pH value of the sample solution played a vital role in influencing the binding purpose on the adsorbent surface during the adsorption process. Figure 8 depicts the adsorption behavior of Ni(II) on the SA-β-CD/CNTs hydrogel adsorbent with respect to pH values ranging from 2 to 6, while the other parameters, such as contact time (120 min), adsorbent dose (1000 mg SA-β-CD/CNTs), and initial concentration (10 mg/L) were kept constant. From the graph, the percentage removal in SA-β-CD and SA-β-CD/CNTs follow a similar trend; however, the two differ in the removal uptake. The percentage removal of Ni(II) on the SA-β-CD hydrogel demonstrates a similar trend to the slope of the percentage removal by SA-β-CD/CNTs. However, the findings found that the percentage removal of Ni(II) on SA-β-CD was lower than SA-β-CD/CNTs. It showed that the Ni(II) removal uptake was more favorable on SA-β-CD/CNTs than SA-β-CD. This finding may be due to additional oxygenated functional groups on CNTs in the SA-β-CD/CNTs hydrogel. At pH 2, the large amount of H^+^ induced a protonation effect on the COO^−^ and OH^−^ groups, causing carboxyl and hydroxyl groups to be favorable for positively-charged development, which is unsuitable for the adsorption process [64]. As a result, the adsorption capacity of Ni(II) in a lower pH environment was reduced proportionally.

Furthermore, when pH increases further, the adsorption capacity is significantly increased, as the protonation effect is weakened. This is because the Ni(II) has the highest binding affinity to the carboxyl and hydroxyl groups on the SA-β-CD and SA-β-CD/CNTs hydrogel adsorbent [63]. However, due to the high concentration of H^+^, particularly at pH 3–4, it continues to obstruct the binding sites of the adsorbent, resulting in a reduced rate of adsorption uptake. The findings demonstrated that the protonation effect was strong and the electrostatic repulsion has been dominated by the positively charged surface of the SA-β-CD/CNTs hydrogel and Ni(II). Nonetheless, at pH 5, with adequate presence of OH^−^ or in slightly alkaline conditions, the removal uptake was easily achieved because Ni(II) was electrostatically attracted to the negative active sites of the SA-β-CD and SA-β-CD/CNTs hydrogel, thus resulting in a rapid decrease in Ni(II) concentration solution [65]. Nevertheless, the adsorption capacity was lessened when the pH solution escalated further to pH 6. This is because the large amount of OH^−^ influenced the electrostatic repulsion between the negative sites of SA-β-CD and SA-β-CD/CNTs adsorbent and Ni(II) metal. It caused strong ionic competition between OH^−^ and Ni^2+^ on the SA-β-CD and SA-β-CD/CNTs adsorbent, resulting in a decrease in adsorption capacity [66,67]. However, the pH study was stopped at pH 6 because a pH above 6 increases the amount of OH^−^ by degrees and influences the development of Ni(II) hydroxides (Ni(OH)_2_) precipitates [68,69]. Therefore, the adsorption capacity study was not tested at pH above 6, and pH 5 was used for further adsorption studies.

#### 3.6.4. Effect of Contact Time and Kinetic Studies

Contact time is a critical parameter that governs adsorption capacity. To illustrate the effect of the exposure period on the removal of Ni(II) metal onto SA-β-CD and SA-β-CD/CNTs, experiments were conducted from 0 to 240 min within 15 min time intervals using retained adsorbent dosage of 1000 mg at 100 mg/L in order to study trends in the adsorption uptake of Ni(II) metal ion. As the relationship between adsorption capacity and contact time can be inferred in Figure 9, the removal efficiency was reported to be rapidly increased in the initial 15 min. The reason for this may be the availability of adsorption penetration sites on the hydrogel adsorbent surface at the beginning of the removal process. An increase in the collision frequency of Ni(II) on the surface of both SA-β-CD and SA-β-CD/CNTs initiates several possible chemical interactions, such as ionic electrostatic interaction between the carboxyl and hydroxyl groups and Ni(II) [70,71]. After further increasing the contact time from 30 to 120 min, the removal uptake gradually increased because most of the penetration sites had been filled. However, it exhibits a steady or plateau curve when the time was further increased from 120 min to 180 min. This indicates that the active penetration sites have reached maxima and been notified that the equilibrium contact time between Ni(II) with both SA-β-CD and SA-β-CD/CNTs has been accomplished. Nevertheless, as the contact time was prolonged to 240 min, the percentage removal was slightly reduced from 62% to 61.3%. This may be caused by the ion–ion repulsion between the Ni(II) ion that was attached to the adsorbent surface and the free distribution of Ni(II) in the solution [72]. In contrast to the percentage removal of SA-β-CD hydrogel, it shows the removal efficiency has reached a plateau curve when prolonged from 120 to 240 min. Based on the experimental data obtained, the percentage removal of Ni(II) on SA-β-CD/CNTs was enhanced compared with SA-β-CD, in which the maximum removal efficiency achieved was about 52.7% and 62%, respectively. Thus, the maximum percentage efficiency and adsorption capacity were achieved at 120 min, which was used for subsequent investigations.

The kinetic adsorption rate is a crucial variable because it controls the time that the Ni(II) species has at the solution interface to migrate on both of the hydrogel adsorbents. Pseudo-first order, pseudo-second order, Bangham and Elovich kinetic models were employed in this work to study the suitability of the adsorption kinetic rate and rate-limiting steps to describe the Ni(II) adsorption process. The kinetic adsorption rate was evaluated through the best-fitted kinetic model, in which a good correlation between the calculated q_e_ value and the experimental q_e_ was recorded, as well as the highest regression coefficient (R^2^) value, shown in Table 1. Based on observations, the experimental data show a good linear relationship between the pseudo-second order kinetics model and the SA-β-CD hydrogel and SA-β-CD/CNTs hydrogel, which provides higher R^2^ values of 0.9993 and 0.9999, respectively. The obtained R^2^ is in good agreement with the pseudo-first order model, Bangham and Elovich. According to previous literature, the highest value of the R^2^ is essential to confirm the best fit kinetic model [73]. In addition, the pseudo-second order kinetic model has shown good agreement with the theoretical and experimental adsorption capacity values. The obtained values were 3.9948 mg/g and 5.9347 mg/g for Ni(II) on SA-β-CD hydrogel and SA-β-CD/CNTs hydrogel, respectively. This revealed that the chemisorption process is the rate-limiting factor in the Ni(II) adsorption reaction on both adsorbents. It presumes that the complexation mechanism or electron transfer between the adsorbent and metal ions have become involved through the chemisorption mechanism [74]. In addition, the equilibrium rate constant of the pseudo-second order sorption (k_2_) was observed to be 0.3883 g/mg min for SA-β-CD hydrogel, and 0.1417 g mg^−1^ min^−1^ for Ni(II) on SA-β-CD/CNTs hydrogel. Hence, the data achieved suggest that the adsorption kinetic process follows a pseudo-second order kinetics model, as illustrated in Figure 10.

#### 3.6.5. Effect of Initial Concentration and Isotherm Studies

Figure 11 shows the effect of the removal efficiency of Ni(II) at various concentrations. The initial concentration of Ni(II) metal ions was varied from 25 mg/L to 100 mg/L onto the SA-β-CD and SA-β-CD/CNTs to evaluate the ability of the adsorbents to capture Ni(II). Based on experimental data, the removal uptake of Ni(II) on both SA-β-CD and SA-β-CD/CNTs rapidly increased from 25 mg/L to 100 mg/L, with a maximum adsorption capacity of 3.81 mg/g and 4.90 mg/g, respectively. The finding shows that SA-β-CD hydrogel and SA-β-CD/CNTs have a greater ability to adsorb Ni(II) when the concentration escalates. This seems to be because of the high ratio of available Ni(II) metal ions to adsorption sites on the adsorbent surface [75]. The percentage removal of Ni(II), SA-β-CD/CNTs is significant, compared with that of SA-β-CD hydrogel. This is because of the high surface area of SA-β-CD/CNTs hydrogel that provides supplemental penetration sites and surface functional groups. Moreover, such conditions might be because the rich surface oxygen-containing functional groups such as hydroxyl and carbonyl increase the electrostatic interaction and improve interactions with Ni(II), which then promotes Ni(II) adsorption on the SA-β-CD/CNTs. Nevertheless, when the initial concentration increased, the percentage removal slope slowly dropped, showing that the penetration sites on the SA-β-CD/CNTs surface had been filled by Ni(II) metal ions [76]. Therefore, 100 mg/L of Ni(II) solution was used as an optimum concentration for further adsorption study.

The adsorption behavior of Ni(II) on SA-β-CD and SA-β-CD/CNTs hydrogel has been thoroughly evaluated using Langmuir, Freundlich, Dubinin–Radushkevich (D–R) and Tempkin isotherm models. The adsorption isotherms model was fitted into a linear form which can be well adopted, within adjustable parameters, by graphical means or by linear regression analysis, in order to investigate the relationship of the experimental data between the concentration of the adsorbed Ni(II) and the adsorption capacity of Ni(II). The corresponding calculated parameters were evaluated, using “Microsoft Excel program” to process the isotherm graphs. The linear adsorption isotherm was considered in this study because the experimental data was more suitable for application than the non-linear form. Moreover, based on the calculated parameters, the coefficient values (R^2^) have been used to determine the best fitted isotherm models for this study, in which the highest value of R^2^ represented the most appropriate isotherm model to describe the adsorption process of Ni(II) to the adsorbent. The calculated parameters for each of the isotherm models are listed in Table 2. For the adsorption mechanism of Ni(II) on SA-β-CD, the results revealed that the adsorption behavior is best fitted to Freundlich isotherm models compared with Langmuir isotherm models. They suggest that the multilayer adsorption of Ni(II) ions has occurred on the surface of hydrogel adsorbents. This can be confirmed by comparing the experimental correlation coefficient (R^2^), where the Freundlich isotherm model achieved a higher R^2^ value compared with Langmuir. Meanwhile, for Ni(II) adsorption on SA-β-CD/CNTs, the best fitted isotherm model based on R^2^ is the Langmuir isotherm model. However, due to the intrinsic parameters of the Langmuir isotherm model, plots gave a negative value of Langmuir constant, b, and the calculated maximum adsorption capacity, Q_m,_ for Ni, as listed in Table 2. The results demonstrate that the Langmuir isotherm model was unsuitable to describe the adsorption process, as the adsorption behavior on the tested system was not following the assumption of the Langmuir isotherm approach [77]. In this context, the Freundlich model was suggested and more fitted to describe the isotherm of the Ni(II) adsorption on SA-β-CD/CNTs hydrogel. Freundlich equilibrium constants were calculated based on the experimental values obtained on the basis of the linear form equation from the plot of ln q_e_ versus ln C_e_, illustrated in Figure 12b,d. Nevertheless, the values of n for Ni on both adsorbents were 1.1528 and 0.08675 L/mg, which lie in the range of 0 < n < 10, showing the adsorption was favorable. The n value indicates the degree of nonlinearity between solution concentration and adsorption as follows: if n = 1, then adsorption is linear; if n < 1, then adsorption is a chemical process; if n > 1, then adsorption is a physical process. The n value in the Freundlich equation was found to be 1.1528 and 0.08675 L/mg on SA-β-CD and SA-β-CD/CNTs, respectively, showing the physical and chemical adsorption conditions involved during adsorption of Ni(II) [76,78].

A Dubinin–Radushkevich (D–R) isotherm was used to study the pseudo-energy of adsorbed adsorbates on the adsorbent surface. The linear equation of DR is expressed as follows:$$\ln\;q_{e}=\ln\;X_{m}-\beta \;\epsilon^{2}$$
$$\epsilon =\mathrm{RT}\;\ln\;\left(1+\frac{1}{\;\mathrm{Ce}\;}\right)$$
where$\;q_{e}$ is adsorbed substances (mg/g), β is the activity coefficient constant at specific energy absorbance, ϵ is Polanyi’s potential, R is the gas constant (J/mol K) and T is the temperature (K). The parameters of D–R were determined by plotting a graph of ln $q_{e}$ versus ϵ^2^, with β obtained from the slope and $X_{m}$ from the intercept [79].

The Tempkin isotherm assumes the adsorption occurs in all molecules of surface of adsorbent and the interaction will be decreased linearly. This model speculates that the relationship between adsorbate–adsorbent that will significantly improve when the heat of adsorption is reduced, subsequently causing the binding energies to be uniformly distributed.
$$q_{e}=\frac{\mathrm{RT}}{b_{t}\;}\;\ln\;A_{T}\;C_{e}$$
$$q_{e}\;=\frac{\mathrm{RT}}{b_{t}\;}\;\ln\;A_{T}+\frac{\mathrm{RT}\;}{b_{t}\;}\;\ln\;C_{e}$$

This equation can be written in another form as follows:$$q_{e}=B_{1}\;\ln\;K_{t}+B_{1}\;\ln\;C_{e}$$
where K_t_ (L/g) is the suitable equilibrium binding constant related to the maximum energy bond, and b_t_ (=RT/βt) is a constant relevant to heat of adsorption (kJ/mol). The slope of $q_{e}$ versus$\;C_{e}$ was used to determine the value of K_t_, and b_t_ was calculated from the intercept of the graph which represents a significant sorbate−sorbent interaction and a minor change in the heat of sorption with temperature change, respectively. $C_{e}$ (mg/g) corresponds to concentration of solution at equilibrium [80].

#### 3.6.6. Effect of Temperature and Thermodynamic

Figure 13 represents the ability of temperature to examine the adsorption performance of Ni(II) metal by SA-β-CD/CNTs hydrogel. A difference in the temperature ranging from 303 K, 323 K and 333 K was demonstrated. The results revealed that a rise in Ni(II) metal solution temperature reduced the potential of both adsorbents to influence positive adsorption capacity. The maximum adsorption capacity of Ni(II) metal reported was up to 5.2158 mg/g and 5.9893 mg/g at 303 K using the prepared SA-β-CD and SA-β-CD/CNTs hydrogel, respectively. The superior absorption performance can be attributed to the high mobility of the Ni(II) metal ion and the opening network structure of the adsorbent when the necessary activation energy is provided [81]. However, with a further increase in the temperature to 333 K, the adsorption efficiency decreased. The result showed that increasing the solution temperature would not increase the number of surface adsorption sites [82]. In addition, a reduction in the percentage removal of Ni(II) at temperature hike might also be due to the domination of the desorption kinetics of the Ni(II) from both SA-β-CD and SA-β-CD/CNTs hydrogels. Hence, this trend suggested overheating the hydrogel resulted in a slight degeneration and incapacitation of the hydrogel adsorbent when the hydrogel was supplied with a more significant external heat [83]. However, based on the effect of temperature, the percentage removal for SA-β-CD/CNTs was improved compared with SA-β-CD hydrogel. To summarize, the optimum temperature for the adsorption performance of Ni(II) has been successfully achieved when the solution temperature is at 303 K.

Moreover, thermodynamic parameters can be obtained from the slope and intercept of the van’t Hoff plot depicted in Figure 13b,c, in which they were calculated according to Equation (7) to (9). The results are summarized in Table 3. Based on the results, the positive value of ΔH° indicates the exothermic nature of the overall Ni(II) adsorption mechanism. In other words, the adsorption of Ni(II) onto SA-β-CD and SA-β-CD/CNTs hydrogel gradually decreases when the temperature rises. Moreover, the negative values of ΔS° might be due to the reduction in the degree of freedom of the system, in which the solid–liquid interface of the adsorbent and Ni(II) solution has not changed significantly. The result obtained is in accordance with the literature reported by Lin et al. [20,82].

### 3.7. Regeneration Studies

Adsorption–desorption studies are another performance indicator of the adsorbent material. Recyclability plays an important role in both the ecological and economic conceivability of various commercial applications. In this work, the reusability potential of the prepared SA-β-CD and SA-β-CD/CNTs hydrogel was assessed by four adsorption–desorption cycles. The adsorption–desorption cycles of prepared SA-β-CD and SA-β-CD/CNTs hydrogel after four consecutive cycles are shown in Figure 14. The adsorption capacities of the hydrogels were slightly reduced after each regenerative cycle for SA-β-CD hydrogel. At the same time, there were no apparent changes in the percentage removal of SA-β-CD/CNTs adsorbent. These results showed a better regeneration potential of SA-β-CD/CNTs hydrogel than SA-β-CD hydrogel for the removal of Ni(II), in which the adsorption efficiency was similar to the first adsorption–desorption cycles where the removal efficiency remains more significant than 60%. Hence, this finding suggested the satisfactory desorption property and reusability of the SA-β-CD/CNTs hydrogel.

### 3.8. Real Water Sample Treatment

The real wastewater samples collected from different sources near to manufacturing industries located in various areas, such as the battery, steel and electronic industries were found suitable to be tested for the removal of Ni(II) ions using the developed hybrid hydrogel adsorbent. Before beginning the removal treatment, the collected wastewater samples were pre-analyzed to quantify the initial amount of Ni(II) in the wastewater sample. Then, the wastewater was spiked with a known concentration of Ni(II) standard solution (100 mg/L). Furthermore, the spiked wastewater samples were then treated with the fabricated SA-β-CD/CNTs hydrogel adsorbent to evaluate the removal uptake efficiency of Ni(II) ions from the real wastewater samples. Table 4 depicts the developed SA-β-CD/CNTs hydrogel adsorbent that has successfully eliminated Ni(II) from the water sample, as is emphasized by the difference in the concentration of the wastewater samples before and after the treatment of the water samples. Moreover, the results have shown that the percentage removal and adsorption capacity of Ni(II) from real wastewater samples were higher than the optimum value obtained. This signifies that the SA-β-CD/CNTs hydrogel adsorbent was seemly superlative in eliminating Ni(II) from the real wastewater environment.

### 3.9. Possible Chemical Interaction of SA-β-CD/CNTs Hydrogel and Ni(II) Metal

Figure 15 depicts the interaction mechanism between SA-β-CD/CNT hydrogel and Ni(II) to study the possible functional groups involved during adsorption procedures. The importance of structural and fundamental studies on the specific elements in the composite has been insinuated by diverse physical and chemical sorption mechanisms that were thoroughly discussed in the experimental results. In this work, chemical interactions such as ion exchange, electrostatic interaction, coordination binding and host–guest interaction were expected to be involved during Ni(II) penetration on SA-β-CD/CNT hydrogel. The interactions mentioned above influenced an essential part of monitoring the adsorption capabilities [83]. Electrostatic interaction behavior was discovered to be the main potential factor that facilitated Ni(II) adsorption onto the economically advanced adsorbent SA-β-CD/CNT hydrogel [20,84,85,86] which are dominated by functional group distribution on raw sorbent materials. In addition, carboxyl groups have pKa values ranging from 1.7 to 4.7, and are deprotonated over an environmentally relevant pH range, allowing heavy metals to interact electrostatically. In contrast, the protonation of hydroxyl groups allows ion exchange interactions with heavy metals at environmentally relevant pH levels [87]. The hydroxyl and carboxyl groups in the SA-β-CD/CNT hydrogel dominated the positive ion of nickel’s adsorption affinity to the SA-β-CD/CNT hydrogel [86]. For instance, the abundance of O-containing groups on alginate, β-CD and CNT that serve as hydrophilic active sites, enable Ni(II) sorption. The OH groups on the outer surface of β-CD can participate in attracting Ni(II) through electrostatic, ion exchange and covalent bonding [53,88]. In addition, Ni(II) ion exchange would possibly necessitate Ca^2+^ displacement from the alginate biopolymer beads during the sorption process [89]. Furthermore, host–guest interaction between β-CD and Ni(II) can be predicted to be involved in the adsorption procedure [82,89]. Moreover, we can conclude that the chemical interaction hydrogel also played an important role in chemical interaction despite the elemental composition, solution chemistry, active surface sites, surface area, porosity and pore volume of adsorption mechanism Ni(II).

## 4. Elemental Composition and Distribution on SA-β-CD/CNTs after Ni(II) Adsorption

The elemental study was carried out using FESEM-EDX spectroscopy to confirm the elemental composition of Ni(II), after the adsorption process onto the surface and into the internal surface of SA-β-CD/CNTs. Figure 16 shows EDXs spectra with three different points used to analyze the Ni(II) sorption properties on each region of the SA-β-CD/CNTs hydrogel. Based on the mean mass of Ni(II) according to the spectra, a mass percentage of Ni(II) of approximately 1.24% was adsorbed. Therefore, the study has proven the adsorption of Ni(II) on SA-β-CD/CNTs from the aqueous environment was successfully justified, in accordance with a study conducted by Çelebi et al. [90].

## 5. Comparative Evaluation of Other Adsorbents

Previous literature has designed various conventional adsorbents as an alternative solution for the removal of Ni(II) from the environment. However, the sorbent showed limited efficiency in the removal of Ni(II) and has still been challenged. This problem is exacerbated by the fact that most of the existing adsorbents are thermally unstable, have poor mechanical properties, are incapable of regeneration, have complicated preparation pathways, have a longer sorption equilibrium time, are expensive and are toxic to the environment [91,92]. Moino et al., found that using individual alginate as biosorption material, extracted from *Sargassum filipendula*, in a dynamic fixed-bed system, resulted in a maximum percentage removal of Ni(II) from industrial wastewater of only 45%. In addition, the authors also mentioned that the prepared alginate had poor potential to regenerate because the desorption after the second cycle was significantly reduced [91]. In contrast, in this study, the removal uptake of Ni(II) reported a greater percentage removal of about 79.86%. Moreover, the prepared SA-β-CD/CNTs hydrogels were excellent and could regenerate for four successive cycles without apparent changes. Hassan et al., studied a novel magnetic responsive alginate/β-cyclodextrin (NTs-FeNPs@Alg/β-CD) polymer bead for the adsorption of heavy metal ions including Ni(II). The authors reported a significant adsorption capacity and removal uptake of Ni(II) onto the prepared adsorbent. However, the NTs-FeNPs@Alg/β-CD adsorbent has a slow sorption and deliberate equilibrium time which requires about 6 h to achieve equilibrium [89], compared with the sorption time of the prepared SA-β-CD/CNTs hydrogels, which demonstrate fast adsorption kinetics of Ni(II); the adsorption was noticed within 15 min of contact, and equilibrium was achieved within 2 h. These results showed an excellent adsorption performance by the prepared SA-β-CD/CNTs hydrogel, due to surface permeability, in this research work. Subsequently, Ahmadi et al., investigated individual modified multi-walled carbon’s capacity for Ni(II) adsorption. The study reported using MWCNTs as adsorbents, which resulted in good Ni(II) adsorption. However, the operation of the adsorption process was restricted during the separation procedure between the solid powder of MWCNTs and Ni(II) solution [91,93]. For this reason, CNTs in this study have been immobilized within a hydrogel–polymer matrix for facile operability [52]. Moreover, another study conducted by Lin et al., using magnetic of Fe_3_O_4_ incorporated into β-CD and multi-walled CNTs for adsorption of Ni(II), showed that the raw material used was not environmentally friendly. This is because, based on Hurbankov et al., the long exposure of Fe_3_O_4_ can lead to respiratory toxicity [20,94,95].

## 6. Conclusions

The prepared SA-β-CD and SA-β-CD/CNTs hydrogel adsorbent were successfully synthesized via graft co-polymerization through an ionic crosslinking reaction and verified by FTIR, FESEM, EDX, TGA and BET. The incorporation of CNTs within the polymer structure of SA-β-CD hydrogel has been proven to be attributable to the improved thermal stability, porosity, and adsorption capacity compared with pristine SA-β-CD hydrogel. The experimental results supported that the complexity of the adsorption system mechanism was fully explained through investigation of solution pH, adsorbent dose, adsorption isotherm models, kinetic models, thermodynamics and regeneration studies. The best adsorption conditions were found when Ni(II) solution concentration was kept at 100 ppm, pH 5, 303 K, and contacted for 120 min and 1000 mg dosage. Hence, the maximum percentage removal of up to 79.86% for SA-β-CD/CNTs and 69.54% for SA-β-CD were reported. The equilibrium adsorption data of Ni(II) on the prepared SA-β-CD and SA-β-CD/CNTs hydrogel matches well with the Freundlich isotherm adsorption isotherm, which illustrates that the Ni(II) adsorption behavior occurred on the multilayer of the adsorbent. The kinetics mechanism was best fitted with a pseudo-second order kinetics model, which signifies the chemisorption mechanism’s involvement. In addition, the chemical interaction mechanism of Ni(II) adsorption on the prepared SA-β-CD and SA-β-CD/CNTs hydrogel involves electrostatic interactions, ion exchange, surface complexation, coordination binding and host–guest interaction. In addition, the prepared hydrogels were investigated for four regeneration cycles and showed that the removal efficiency of Ni(II) on SA-β-CD hydrogel was gradually depleted. Meanwhile, the Ni(II) removal efficiency of the SA-β-CD/CNTs hydrogel was maintained without apparent decrease. The result indicates that SA-β-CD/CNTs hydrogel could be reused multiple times to adsorb wastewater pollutants efficiently; thus, it can become a definitive alternative adsorbent in future large-scale wastewater treatment.

## Figures and Tables

**Figure 1 polymers-14-05524-f001:**
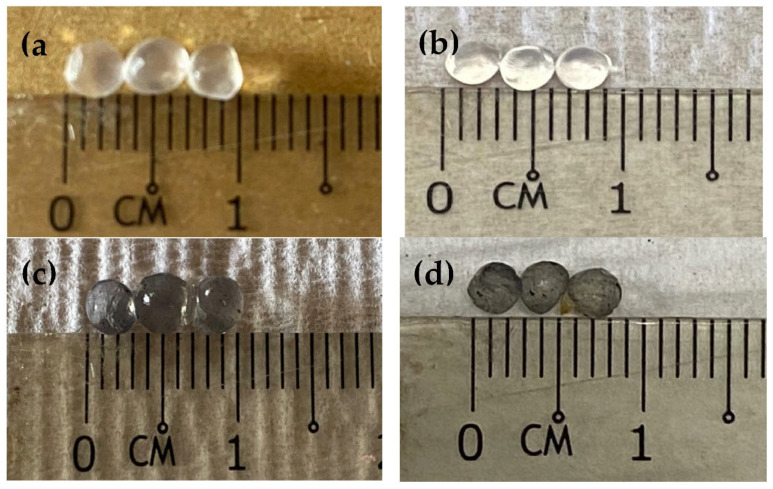
Physical image of the (**a**) swollen SA−β−CD hydrogel (**b**) air-dried hydrogel SA−β−CD (**c**) swollen SA−β−CD/CNTs hydrogel (**d**) air-dried hydrogel SA-β-CD/CNTs.

**Figure 2 polymers-14-05524-f002:**
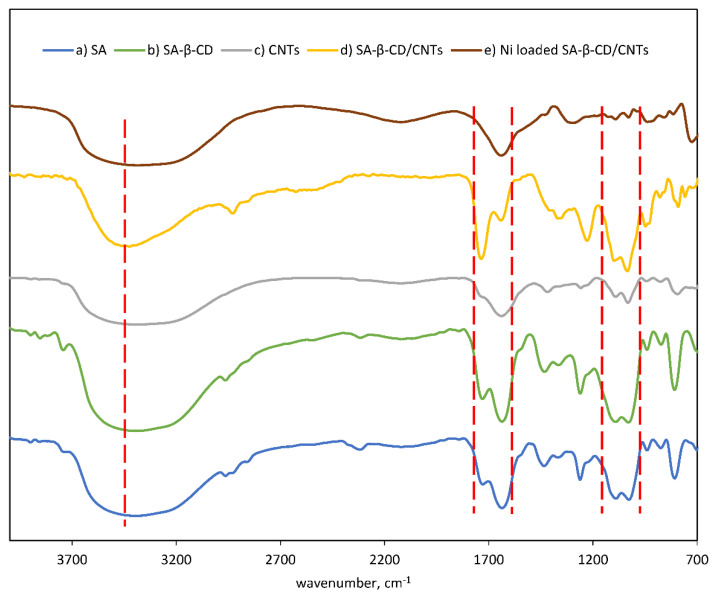
FTIR spectra of (**a**) SA hydrogel (**b**) SA−β−CD (**c**) CNTs (**d**) SA−β−CD/CNTs (**e**) Ni loaded SA−β−CD/CNTs to study the effect of incorporation of CNTs into hybrid hydrogel.

**Figure 3 polymers-14-05524-f003:**
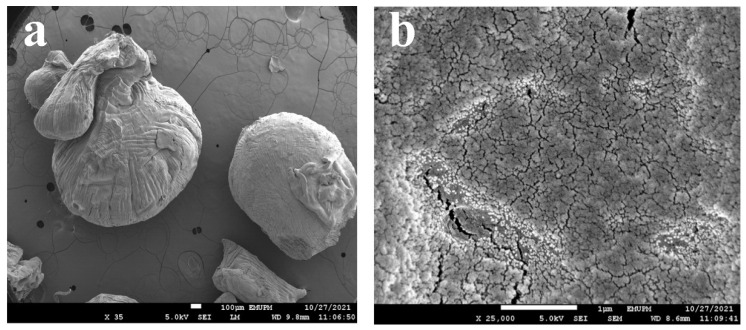
FESEM images of SA−β−CD/CNTs before adsorption at different magnifications of (**a**) 35× (**b**) 25,000×.

**Figure 4 polymers-14-05524-f004:**
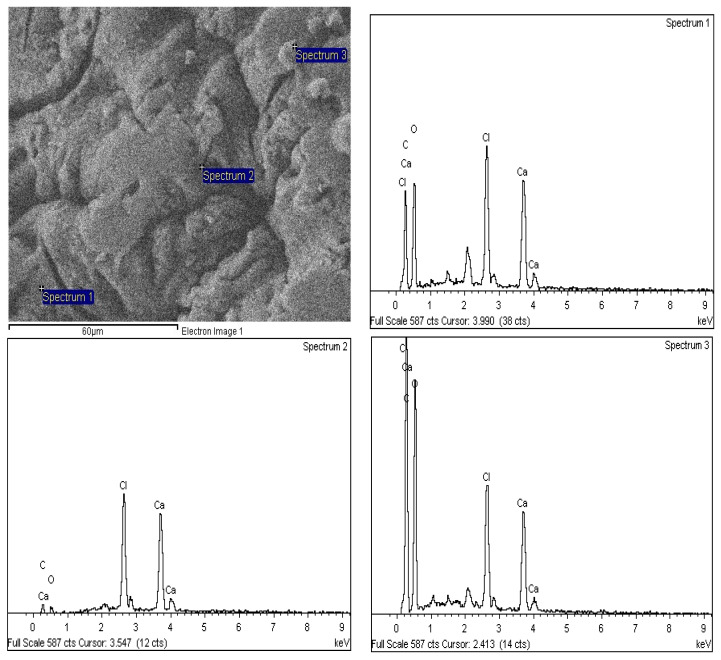
EDXs spectra for SA−β−CD/CNTs hybrid hydrogel at three different regions.

**Figure 5 polymers-14-05524-f005:**
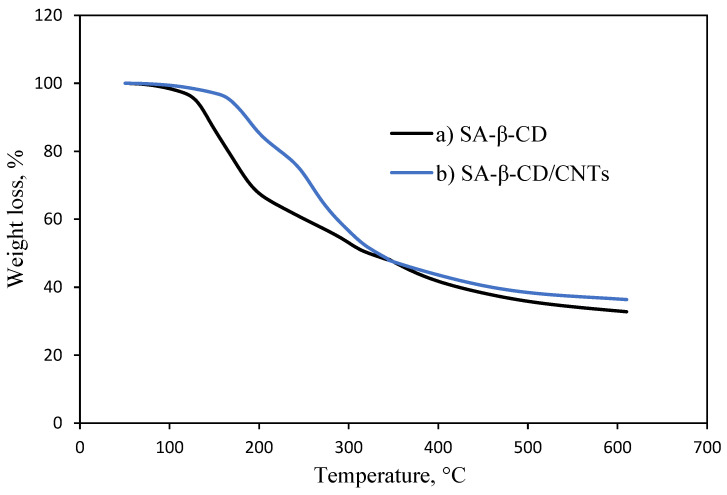
Thermogravimetric analysis of (a) SA−β−CD and (b) SA−β−CD/CNTs.

**Figure 6 polymers-14-05524-f006:**
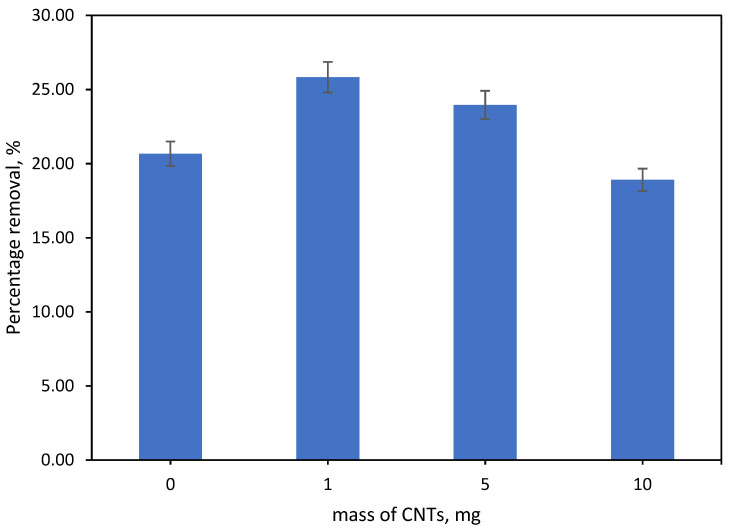
Effect mass of incorporated CNTs. Condition: [Ni(II)] = 10 mg/L, pH= 4, adsorbent dosage = 500 mg, contact time = 60 min.

**Figure 7 polymers-14-05524-f007:**
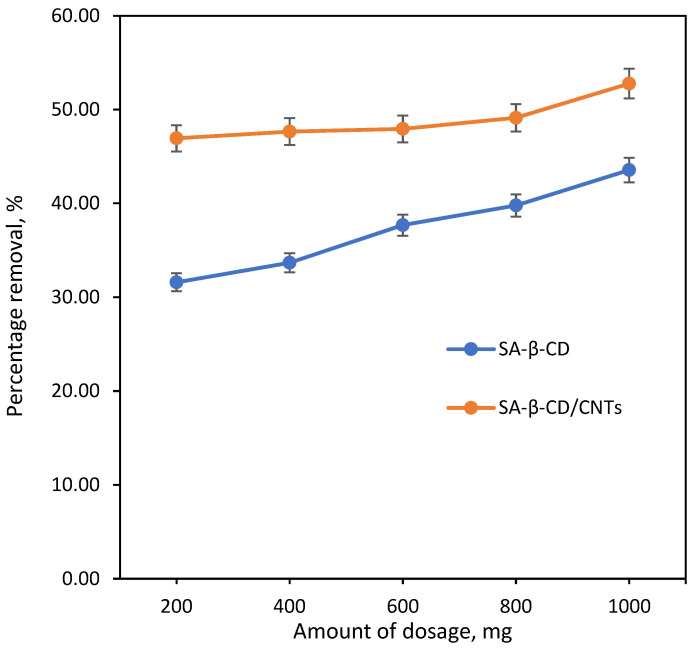
Effect of dosage condition: [Ni(II)] = 10 mg/L, pH = 4, contact time = 60 min.

**Figure 8 polymers-14-05524-f008:**
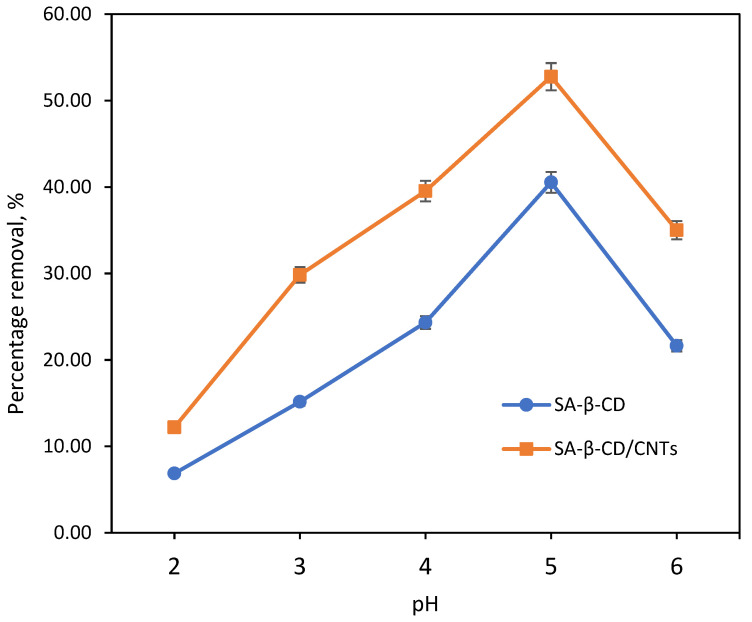
Effect of pH. Condition: [Ni(II)] = 10 mg/L, dosage = 1000 mg, contact time = 60 min.

**Figure 9 polymers-14-05524-f009:**
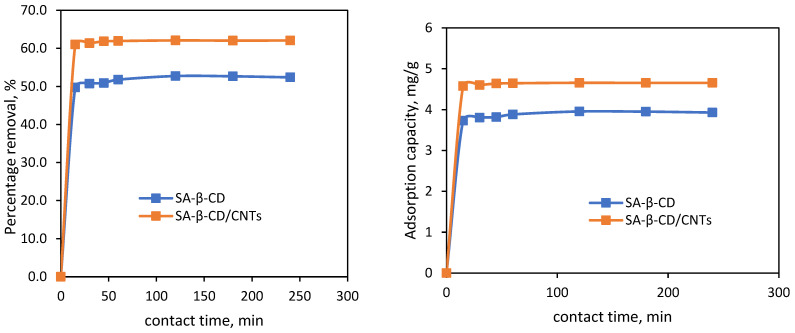
Effect of contact time on Ni(II) adsorption SA−β−CD hydrogel and SA−β−CD/CNTs hydrogel.

**Figure 10 polymers-14-05524-f010:**
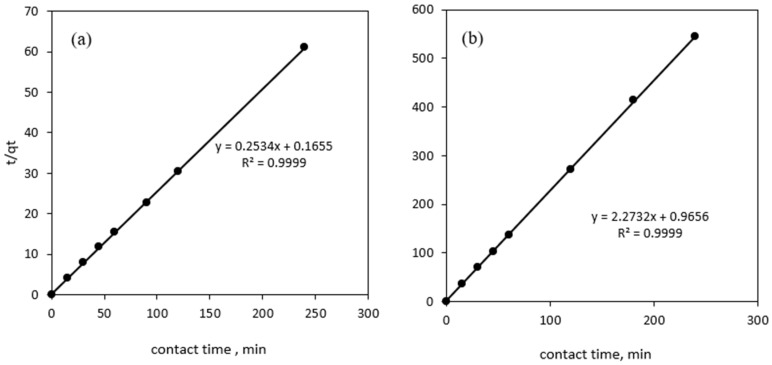
Plot of (**a**) Pseudo second-order kinetic plot on SA-β-CD hydrogel (**b**) Pseudo second-order kinetic plot on SA-β-CD/CNTs hydrogel of Ni(II) adsorption.

**Figure 11 polymers-14-05524-f011:**
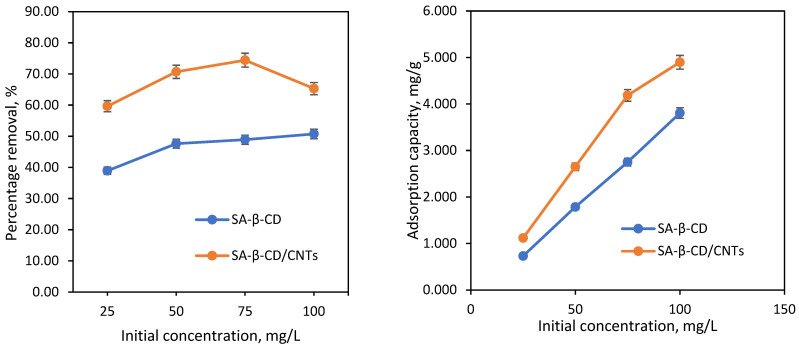
Effect of initial concentration. Condition: [dosage = 1000 mg, contact time = 120 min, pH = 5].

**Figure 12 polymers-14-05524-f012:**
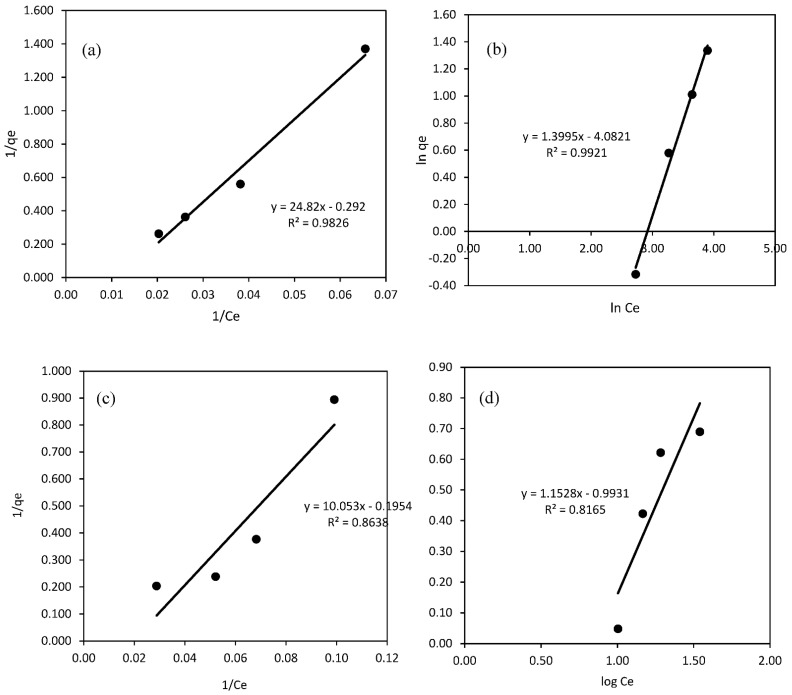
Isotherm plot of (**a**) Langmuir model for SA−β−CD hydrogel (**b**) Freundlich model for SA−β−CD hydrogel (**c**) Langmuir model for SA−β−CD/CNTs hydrogel (**d**) Freundlich model for SA−β−CD/CNTs hydrogel.

**Figure 13 polymers-14-05524-f013:**
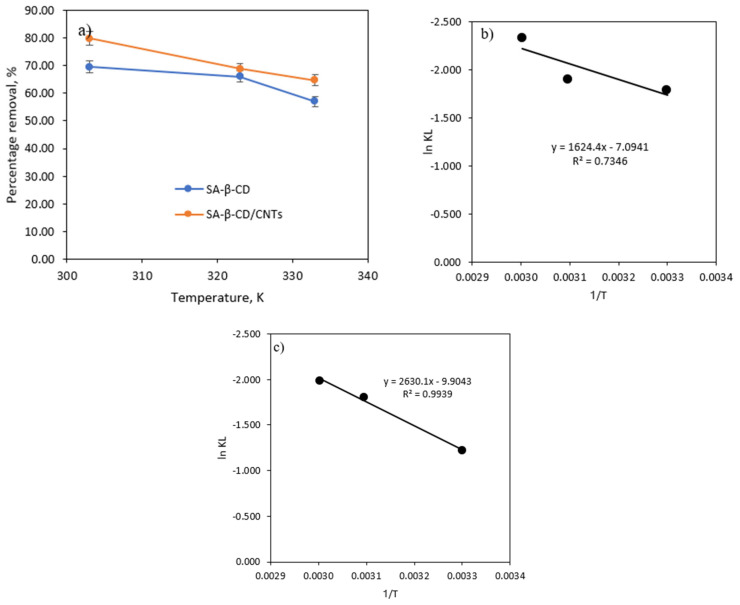
(**a**) Effect of temperature of Ni(II) on SA−β−CD/CNTs hydrogel (**b**) van’t Hoff plot for SA−β−CD (**c**) van’t Hoff plot for SA−β−CD/CNTs.

**Figure 14 polymers-14-05524-f014:**
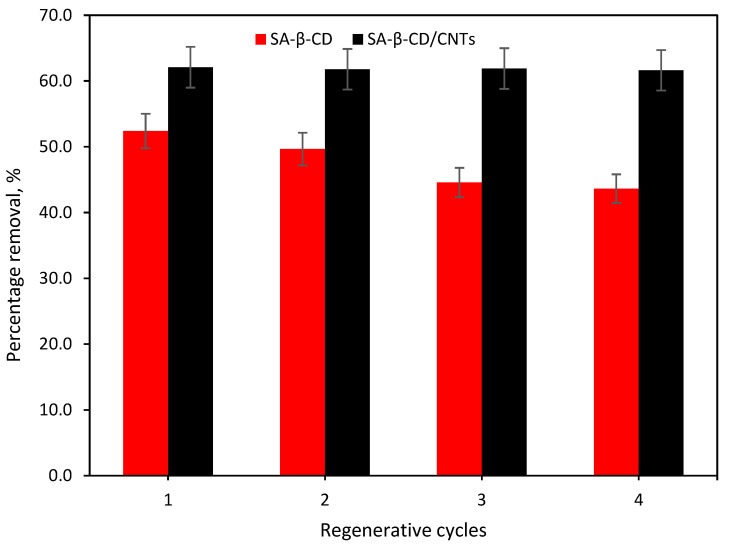
Regeneration cycles of SA−β−CD hydrogel and SA−β−CD/CNTs hydrogel adsorbent.

**Figure 15 polymers-14-05524-f015:**
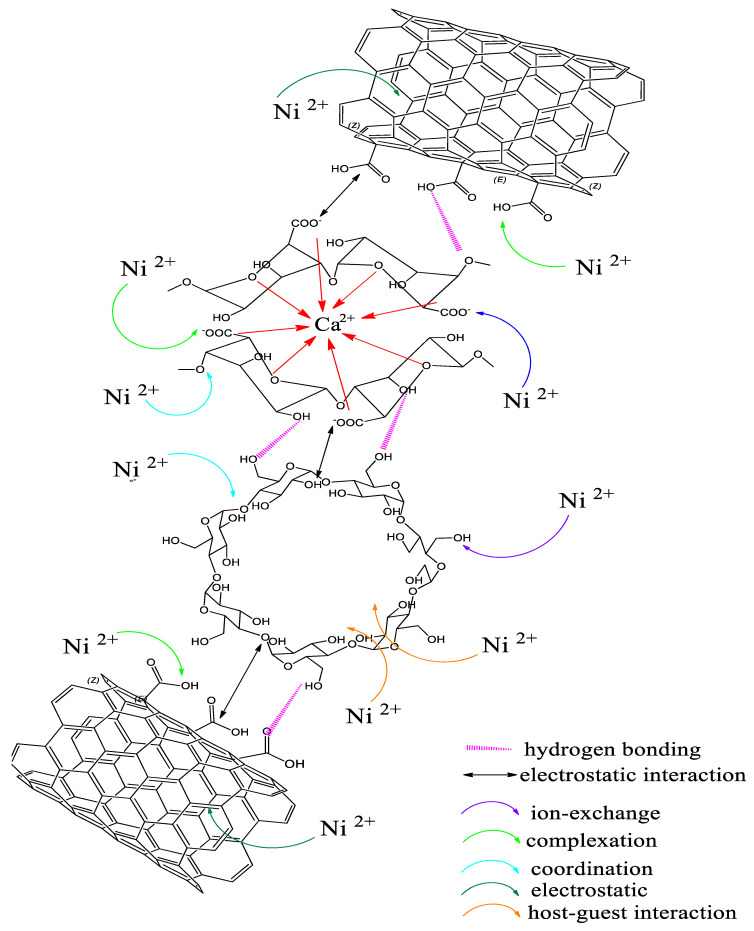
Probable schematic interaction mechanism of the SA−β−CD/CNTs hydrogel adsorbent.

**Figure 16 polymers-14-05524-f016:**
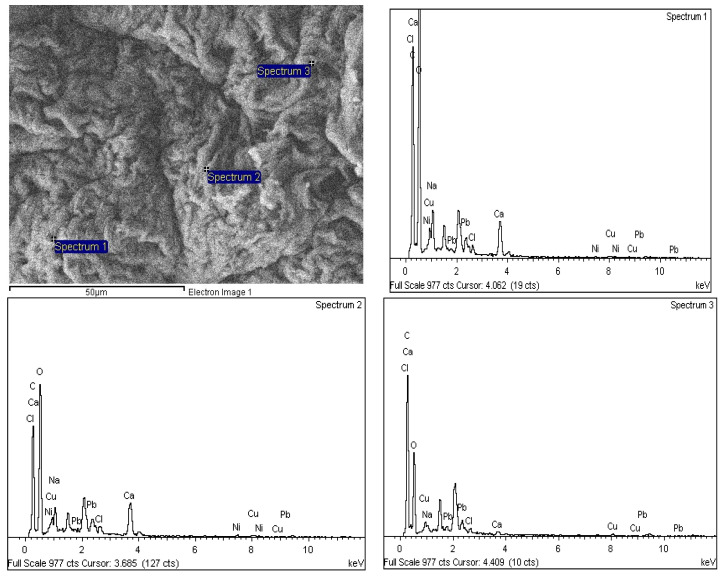
EDX spectra of SA−β−CD/CNTs hydrogel after Ni(II) adsorption.

**Table 1 polymers-14-05524-t001:** Kinetics parameters.

Kinetic Modelling	Parameters	Type of Adsorbent	
		SA-β-CD	SA-β-CD/CNTs
Pseudo-first order	k_1_ (1/min)	0.00002	0.00009
	q_e_ (mg/g)	3.7396	2.8933
	R^2^	0.65717	0.51638
Pseudo-second order	k_2_ (g/mg min)	0.3883	0.1417
	q_e_ (mg/g)	3.9948	5.9347
	R^2^	0.9993	0.999
Bangham	K_o_	0.0215	0.0062
	α	0.0906	0.1162
	R^2^	0.8962	0.80782
Elovich	α	0.3384	0.3244
	β	0.0905	0.0347
	R^2^	0.8962	0.8086

**Table 2 polymers-14-05524-t002:** Isotherm parameters.

Isotherm	Isotherm Parameter	Type of Adsorbent	
		SA−β−CD	SA−β−CD/CNTs
Langmuir	q_max_	−3.4247	−18.7970
	b (L/mg)	−0.0118	−0.0082
	R^2^	0.9826	0.8591
Freundlich	K_f_	0.0169	0.1016
	n	1.1528	0.8675
	R^2^	0.9921	0.8165
Tempkin	K_t_	0.0826	1.6040
	B_t_	2.5473	3.0770
	R^2^	0.9732	0.9045
D–R	B	−0.00006	−0.00005
	q_max_	5.2179	5.9960
	R^2^	0.9728	0.9861

**Table 3 polymers-14-05524-t003:** Thermodynamic parameters.

Adsorbent	Temperature (K)	ΔG° (kJ/mol)	ΔH° (kJ/mol)	ΔS° (kJ/mol K)
SA-β-CD	303	4.4970	−13.5053	−0.0590
	323	5.0954		
	333	6.4520		
SA-β-CD/CNTs	303	3.056	−21.8670	−0.0823
	323	4.826		
	333	5.486		

**Table 4 polymers-14-05524-t004:** Analysis of the real wastewater samples before and after treatment.

Wastewater Industry	Before Treatment, C_i_ (mg/L)	After Treatment, C_e_ (mg/L)	Q_e_ (mg/g)	% Removal
Battery	100	35.94	4.8045	64.06
Electronic	100	34.28	4.9290	65.72
Steel	100	33.98	4.9515	66.02

## Data Availability

The data presented in this study are available on request from the corresponding author.

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
