# Peer review of "Sodium Alginate/β-Cyclodextrin Reinforced Carbon Nanotubes Hydrogel as Alternative Adsorbent for Nickel(II) Metal Ion Removal"

_polymers, 2022, doi:10.3390/polym14245524_

Round 1

Reviewer 1 Report

Comments: The topic is interesting. However there are many issues should to be addressed 

1-    There are many English typos errors should have been revised strongly over all the text.

2-    Authors didn’t discuss deeply  why CNTs were used and if they can isolate CNTs after adsorption to confirm presence NI(II) inside their cavity?

3-    CNTs were not exactly described if they are multi-walled or single walled and how can they be oxidized?

4-    Statistical analysis  should to be used to observe the difference between swelling  hydrogel beads and dried hydrogel beads. the experiment should to be carried by weight.

5-    Thin ultrastructure longitudinal edge should have been used  to investigate inner face (internal structure) of beads

6-    CNTs were not included in FTIR spectra. Authors have to investigate their spectrum after oxidation.

7-    EDS mapping SEM images have to be used to illustrate  distribution of  absorbed NI(II) in hydrogel.

8-    Authors should to design scheme to illustrate the chemical interaction and crosslinking behaviour of hydrogel

9-    REFs should to be revised according to polymer style format journal

Author Response

Thank you for your thoroughly read this manuscript. I hope you will consider this manuscript for publication to Polymers. 

Author Response

Thank you for your thoroughly reading of this manuscript. I hope you will consider the manuscript for publication to Polymers. 

Reviewer 3 Report

Manuscript Number: Polymers-2069021

Title: Sodium Alginate/β-Cyclodextrin Reinforced Carbon Nano-tubes Hydrogel as Alternative Adsorbent for Nickel (II) Metal Ion Removal

The manuscript deals with the preparation of sodium alginate/β-cyclodextrin beads reinforced by different percentages of carbon nanotubes and then utilized for NI (II) ion adsorption. I would like to accept this paper for the publication in POLYMERS after clarifying the following comments. 

  • First of all, you have to compare your SA-β-CD/CNTs results with SA-β-CD then only anyone can understand how much improvement you did in your work.
  • Your experimental Qmax and from Langmuir (In table 2) of SA-β-CD/CNTs is too low? Then how it is best, Explain?
  • The authors used CNTs that don’t contain any functional groups, therefore what is the use of reinforced CNTs in this paper explain? 
  • The authors should improve the introduction part based on CNTs based materials, how it improves the adsorption performance?
  • The authors did not present any swelling studies, which is very crucial for any adsorption studies. Further it’s difficult to estimate the absorption capacity of SA-β-CD/CNTs hydrogel beads with pristine SA-β-CD hydrogel beads?
  • Therefore, please provide the swelling studies in water and at different pHs?
  • Authors should use the codes/notations for different CNTs concentrations and use the best one throughout the manuscript?
  • Provide the pristine SA-β-CD hydrogel beads image in figure 1, for better understanding. 
  • Please mark the changes in the FTIR graph to better understand?
  • The authors don’t mention which surface morphology of SA-β-CD/CNTs sample code presented? Also stated porous structure in section 3.3, please provide the pore size? Additionally, provide the surface morphology of pristine SA-β-CD hydrogel beads, and compare its morphology with SA-β-CD/CNTs?
  • Similarly, provide the EDS analysis for the pristine SA-β-CD hydrogel beads?
  • The authors compared Ni(II) ion adsorption by changing CNTs in SA-β-CD hydrogel beads only but they did not compare with pristine (SA-β-CD) beads, then how did you confirm the SA-β-CD/CNTs beads is best for adsorption? What’s the useful point in this study? 
  • In figure 6, the authors varied CNTs and used for adsorption of Ni(II) ion;

Why don't you use the 75 mg/L solution for this study? Further, the obtained results show that the adsorption capacity is very low?

  • Provide the labels for figure 9, such as a) and b)
  • Similarly, provide the labels for figure 11, and further a) is having borders with blue but not b) is with black borders
  • The temperature effect mentioned in figure 12, however, you can use the thermodynamics equation (by plotting a graph) for a better explanation regarding the adsorption?
  • The authors should compare the regeneration cycles of adsorbent SA-β-CD/CNTs with pristine SA-β-CD beads?
  • The authors should re-write both the abstract and conclusions according to results.
  • Finally authors should compare their results with other literature such as alginate or alginate/β-CD or CNTs based materials and explain how these results are better than others?

Author Response

Thank you for your careful and thoroughly reading of this manuscript. I hope you will consider the manuscript for suitable publication to Polymers. 

Round 2

Reviewer 1 Report

Manuscript was revised point by point according to reviewer comments and it can be accepted in present from.

Author Response

We would like to thank the reviewer for careful and thorough reading of this manuscript and for the thoughtful comments and constructive suggestions, which help to improve the quality of this manuscript.

Reviewer 2 Report

1-      The manuscript still suffers from grammatical errors.

2-      On P. 4, the authors did not cite the Mohammadinezhad et al. paper.

3-      On the same page, the authors cited ref. #21 as Lin et al., while Lin et al. is stated as 20.

4-      The authors did not write subscripts properly.

5-      Both parts of my previous #3 comment have not been applied. The unit style is not consistent throughout the manuscript.

6-      The authors cited ref #60 for effect of CNT, however, that ref. does not even involve CNT.

Author Response

We would like to thank the reviewer for careful and thorough reading of this manuscript and for the thoughtful comments and constructive suggestions, which help to improve the quality of this manuscript. Our response to your comment in the attachment. 

Reviewer 3 Report

Thank you for your response.

I recommend this manuscript to be published in Polymers 

Author Response

(The authors gave the same response as above.)

Round 3

Reviewer 2 Report

Thanks the authors for responding to the comments. The manuscript can now be published in Polymers.